# Genome-resolved metagenomics suggests a mutualistic relationship between *Mycoplasma* and salmonid hosts

Jacob A. Rasmussen [1,2✉], Kasper R. Villumsen[3], David A. Duchêne[2], Lara C. Puetz[2], Tom O. Delmont[2,4], Harald Sveier[5], Louise von Gersdorff Jørgensen[6], Kim Præbel[7], Michael D. Martin[8], Anders M. Bojesen[3], M. Thomas P. Gilbert [2,8], Karsten Kristiansen [1,9] & Morten T. Limborg [1,2✉]

Salmonids are important sources of protein for a large proportion of the human population. *Mycoplasma* species are a major constituent of the gut microbiota of salmonids, often representing the majority of microbiota. Despite the frequent reported dominance of salmonid-related *Mycoplasma* species, little is known about the phylogenomic placement, functions and potential evolutionary relationships with their salmonid hosts. In this study, we utilise 2.9 billion metagenomic reads generated from 12 samples from three different salmonid host species to I) characterise and curate the first metagenome-assembled genomes (MAGs) of *Mycoplasma* dominating the intestines of three different salmonid species, II) establish the phylogeny of these salmonid candidate *Mycoplasma* species, III) perform a comprehensive pangenomic analysis of *Mycoplasma*, IV) decipher the putative functionalities of the salmonid MAGs and reveal specific functions expected to benefit the host. Our data provide a basis for future studies examining the composition and function of the salmonid microbiota.

[1] Laboratory of Genomics and Molecular Medicine, Department of Biology, University of Copenhagen, Copenhagen, Denmark. [2] Center for Evolutionary Hologenomics, GLOBE institute, Faculty of Health and Medical Sciences, University of Copenhagen, Copenhagen, Denmark. [3] Department of Veterinary and Animal Sciences, University of Copenhagen, Veterinary Clinical Microbiology, Copenhagen, Denmark. [4] Génomique Métabolique, Genoscope, Institut François Jacob, CEA, CNRS, Univ Evry, Université Paris-Saclay, Evry, France. [5] Lerøy Seafood Group ASA, Bergen, Norway. [6] Department of Veterinary and Animal Sciences, University of Copenhagen, Parasitology and Aquatic Pathobiology, Copenhagen, Denmark. [7] Norwegian College of Fishery Science, UiT the Arctic University of Norway, Tromsø, Norway. [8] Department of Natural History, NTNU University Museum, Norwegian University of Science and Technology (NTNU), Trondheim, Norway. [9] Institute of Metagenomics, BGI-Shenzhen, Shenzhen, China. ✉email: jacob.rasmussen@bio.ku.dk; morten.limborg@sund.ku.dk

The microbial communities that inhabit the vertebrate gastrointestinal tract are tightly connected to many traits displayed by the host[1–4]. Nonetheless, we are far from understanding the ecological and evolutionary forces that structure these microbial communities. Microbial generalists have been investigated in teleosts to decipher these ecological forces[5]. Recently, several reports using 16S rRNA gene amplicon surveys of the gut microbiome in salmonids have revealed that the dominant bacterium belongs to the genus *Mycoplasma*, and that this bacterium was unknown until recently[6–8]. These salmonid related *Mycoplasma* species have been shown to be highly dominant in the gastrointestinal microbiota of all salmonids investigated, including rainbow trout (*Oncorhynchus mykiss*)[6,9,10], Chinook salmon (*Oncorhynchus tshawytscha*)[7] and Atlantic salmon (*Salmo salar*)[11–13]. Further, phenotypic evidence now points to an advantageous role of the abundant *Mycoplasma* including increased disease resiliency revealed by a striking inverse correlation between the abundance of *Mycoplasma* and a potentially pathogenic *Vibrio sp.*[6].

*Mycoplasma* is one of the smallest independently self-replicating organisms known. The small genome size of *Mycoplasma* spp. is hypothesised to be a result of close interaction with their host, which has resulted in subsequent gene loss[14,15]. *Mycoplasma* species are bacteria lacking a cell wall and are in general characterised by small physical dimensions and a small genome. Most host-associated *Mycoplasma* genomes currently sequenced are smaller than 1 Mb and contain fewer than 1000 protein-encoding genes[16–18]. Especially because of the small genome size, this genus of Mollicutes has one of the most extensive compilations of genomic sequences[19]. *Mycoplasmas* are recognised as either parasitic or commensals to their host and have undergone a reductive evolution from the *Bacillus/Clostridium* branch of Gram-positive eubacteria[18], which often results in a reduction of the number of genes in the genome. Despite the reduction of genes, *Mycoplasma* displays a vast variety of phenotypic characteristics, like adaptations to their host, pathogenesis, and mobility[18].

Despite the observed dominance of *Mycoplasma* in salmonid hosts, very little is known about *Mycoplasma* and its functional potential in salmonids and other teleosts[20]. In this study (I) we present the first high-quality metagenome-assembled genomes (MAGs) of *Mycoplasma* from multiple salmonid species including shotgun sequencing of intestinal content from rainbow trout, Atlantic salmon, and European whitefish (*Coregonus lavaretus*), (II) we establish the phylogeny of the salmonid associated MAGs based on a comparison with 44 known *Mycoplasma* genomes, leading to the identification of putative novel *Mycoplasma* species, (III) we present a comparative genomics analysis of *Mycoplasma* to study their functionality in salmonids in the context of other species within the same genus. Lastly, (IV) we examine functions of the salmonid MAGs that may be related to a potential adaptation to the intestinal host environment.

## Results

**Genome resolved metagenomics reveals dominance of *Mycoplasma* in the gut microbiota across three salmonid species.** We sampled gut content from three salmonid species from different environments. Two species relevant for aquaculture were chosen, including juvenile rainbow trout and adult Atlantic salmon. Juvenile rainbow trout were sampled in a land-based, freshwater recycled aquaculture system (RAS) in northern Denmark. Atlantic salmon were sampled from a commercial cohort in an open water net pen near Bergen, Norway. Thirdly, wild European whitefish were sampled from a freshwater lake in northern Norway (Fig. 1a) to represent a phylogenetic outlier for comparison (Fig. 1a).

A total of 2.9 billion metagenomic reads were generated from 12 individuals. Of those, 297 million reads passed quality control and host filtering criteria. Estimates of saturation revealed a sufficient sequencing depth for rainbow trout, but not fully saturated for Atlantic salmon and European whitefish (Supplementary Fig. 1). These filtered reads represent the gut microbiome of the three hosts studied here and were used to establish three metagenomic co-assemblies (Supplementary Data 1). A combination of automatic and manual binning was applied to each co-assembly output, which resulted in one manually curated, low redundant and a near-complete metagenome-assembled genome (MAG), related to *Mycoplasma*, from each host species (Table 1) (Supplementary Fig. 2). Though sequencing depth was not fully saturated for Atlantic salmon and European whitefish, we do not suspect this to have a major impact on *Mycoplasma* MAGs, since they all were of a high completion (Table 1).

Metagenomic data from gut content of juvenile rainbow trout, adult Atlantic salmon, and European whitefish revealed a high relative abundance of *Mycoplasma*. The salmonid MAGs comprised $\bar{x} = 69.44\%$ (SD ± 11.45%) of the metagenomic reads in rainbow trout, $\bar{x} = 72.98\%$ (SD ± 13.07%) of the metagenomic reads in Atlantic salmon, and 58.03% of the metagenomic reads in European whitefish (Fig. 1b). Investigation of bacterial load in juvenile rainbow trout, using real-time polymerase chain reaction

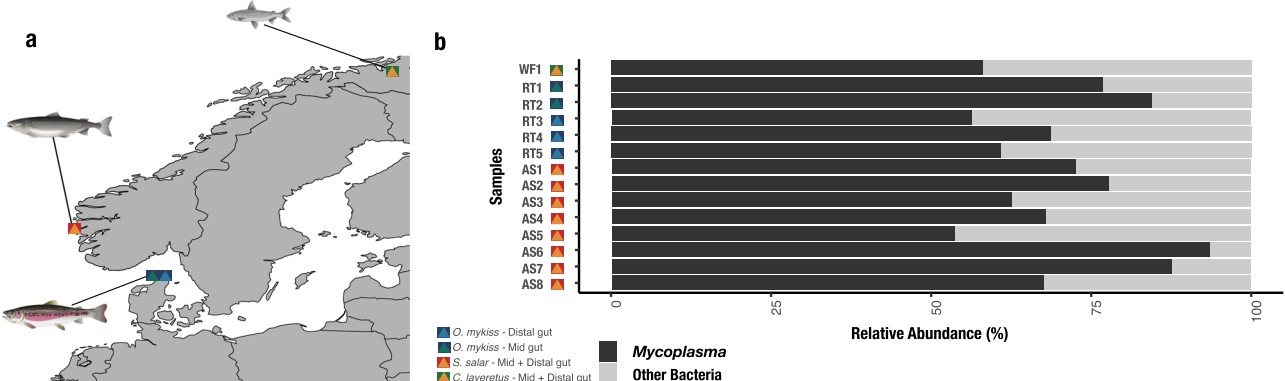

**Fig. 1 Sample overview. a** Map of Scandinavia, where sampling points are indicated. Host and sample types are specified in the legend. The European whitefish sample is shown as a yellow triangle with a green square, Atlantic salmon samples are illustrated as a yellow triangle with a red square. Rainbow trout are visualised with two different sample types for the mid and distal gut, with green and light blue triangles, respectively. Both sample types have a dark blue square. **b** Relative abundance of *Mycoplasma* across all samples. The black bar plot indicates the presence of *Mycoplasma*, whereas all other bacteria are concatenated as "other bacteria".

**Table 1 Summary of metagenomic assembled genomes (MAGs) related to *Mycoplasma* in three salmonid hosts.**

| Host species | No. of contigs | N50 (bp) | Total length (bp) | GC content (%) | CheckM completion (%) | CheckM contamination (%) | No. of genes in the genome |
|---|---|---|---|---|---|---|---|
| Atlantic Salmon | 243 | 3598 | 660,624 | 26.1 | 86.24 | 1.76 | 844 |
| Rainbow Trout | 35 | 88,883 | 697,833 | 24.8 | 95.00 | 0.38 | 627 |
| European Whitefish | 66 | 27,905 | 701,798 | 26.4 | 99.62 | 0.25 | 664 |

(PCR) of the 16S rRNA gene, revealed an average cycle threshold (CT) value of 36.3 (SD ± 2.8), where water samples from RAS resulted in an average CT value of 27.6 (SD ± 2.8), clearly indicating a lower bacterial load in the intestinal environment compared to the surrounding water. The amount of sequencing effort to obtain any microbial data and qPCR results of the 16S rRNA gene indicates low bacterial biomass in juvenile rainbow trout gut content samples[21] (Supplementary Fig. 1 and Supplementary Table 1).

The GC content of the salmonid MAGs was in the lower end compared to the other *Mycoplasma* species included (Table 1, Fig. 3). Previous investigations of Tenericutes' genomes reported GC content varying from 20 to 70%, indicating high plasticity[22]. Initial analysis of average nucleotide identity (ANI) clustering and gene clusters from each sample to investigate strain variation within each sample revealed identical MAGs within each respective host and clustered each individual salmonid MAGs according to host species (Supplementary Fig. 3a).

We used fluorescence microscopy to visualise the presence of bacteria on the distal gut epithelial surface as this would further indicate a functional adaptation of the bacteria to the intestinal environment of juvenile rainbow trout. The identity of bacteria could not be established using specific probes, which is likely due to the low bacterial biomass impeding the generation of clear signals. An alternative explanation is the lack of positive control for probe design; unfortunately, no appropriate reference exists since it has yet not been possible to culture these salmonid associated Mycoplasma species[20]. However, our investigation did reveal a clear DAPI based signal from bacterial cells in close contact with the rainbow trout epithelial surface (Supplementary Fig. 4a–d), which we hypothesise are likely to include *Mycoplasma* cells, based on the observation that >50% of all microbial reads belonged to *Mycoplasma* (Fig. 1b).

**Phylogenomics reveal candidate species of *Mycoplasma* in salmonids**. We performed a phylogenomic analysis to place the salmonid MAGs within *Mycoplasma*. To do so, we generated a database of protein clusters across 44 genomes of *Mycoplasma* isolated from multiple tissue types and host species (Supplementary Data 1). Gene annotation using Hidden Markov Models (HMMs) resulted in 2646 hits, and data filtering for phylogenomics led to a final data set of 55 single-copy core gene alignments from 50 genomes (includes representatives of *Ureaplasma* and *Bacillus*).

Phylogenetic analyses consistently recovered several highly supported groupings in the genus *Mycoplasma* and led to a robust placement of our three salmonid MAGs. Two of the salmonid MAGs isolated from the fish subfamily Salmoninae, species rainbow trout and Atlantic salmon, clustered together with an ANI of 96.1% and formed a monophyletic group with the grouping of *Mycoplasma penetrans* and *Mycoplasma iowae* (Fig. 2 and Supplementary Fig. 5a), the latter being commonly found in the intestine of turkey (*Meleagris gallopavo*)[23]. The two Salmoninae MAGs likely belong to the same species, as suggested by their short terminal branches and uniquely strong branch support according to multiple metrics. The fact that their ANI

relative to *M. penetrans* and *M. iowae* was <80% further indicated that the two MAGs correspond to, to our knowledge, a new salmonid related *Mycoplasma* species (Fig. 3). The close relationship of these salmonid MAGs also indicates that they have a close ecological association with Salmoninae, rather than originating from the environment surrounding the host (Fig. 2 and Supplementary Fig. 5a).

The third salmonid MAG, characterised by the European whitefish, subfamily Coregoninae, was not identified as a close relative of the isolates from Salmoninae (Fig. 2; Supplementary Fig. 5a). Rather this salmonid MAG appears to be distinct from any of the reference *Mycoplasma* species, with the closest relative found to be *Mycoplasma mobile* (ANI < 80%), a pathogen isolated from the gills of tench (*Tinca tinca*) (Fig. 3).

Overall, our analysis indicated that the salmonid MAGs we characterised represent two salmonid related *Mycoplasma* species. We tentatively name them according to their respective host species: *Candidatus* M. salmoninae and *Candidatus* M. lavaretus. Furthermore, we divided *Candidatus* M. salmoninae into two biotypes according to their salmonid host, rainbow trout and Atlantic salmon, resulting in *Candidatus* M. salmoninae mykiss (MSM), *Candidatus* M. salmoninae salar (MSS) and *Candidatus* M. lavaretus (ML), respectively.

Beyond the scope of Salmonid MAGs, the phylogenomic analyses also recovered several clades of the genus *Mycoplasma* with high confidence according to most metrics of phylogenetic branch support (Fig. 2 and Supplementary Fig. 3a, b), confirming the previous phylogenetic placement of Mycoplasma[22,24]. One metric of branch support was low for most branches, internode certainty for sites (sIC), reflecting the fast accumulation of substitutions in the genus. A fast-evolutionary rate relative to the taxonomic scale of the data is also reflected in the saturation of substitutions found in 12 genes (consequently excluded from phylogenetic analyses), and in the long estimated terminal branches across samples.

**An open pangenome with diverse sets of functions is in accordance with niche adaptations of *Mycoplasma*.** We performed a comparative analysis to place the salmonid MAGs within *Mycoplasma* with respect to their gene content and gene functions. Our *Mycoplasma* pangenome included 37,158 open reading frames (ORFs) from the 44 different reference *Mycoplasma* genomes, two *Ureaplasma* genomes and the three salmonid MAGs, and identified a total of 18,021 gene clusters. Surprisingly, the single-copy core genes which were present among all genomes of *Mycoplasma* and *Ureaplasma* genomes only comprised 1.34% of the total ORFs of the pangenome with 10 gene clusters and 499 ORFs (Fig. 3). The amount of singleton gene clusters in the pangenome (i.e., gene clusters only present in a single genome) was 62.8% of all the gene clusters. We investigated the openness of the pangenome using Heaps' law, resulting in $\alpha = 0.281$, confirming an open pangenome[25], indicating specific adaptations with accessory genes and gene losses, which could derive from specific niche adaptations, like specific host environments and symbiosis (Fig. 3).

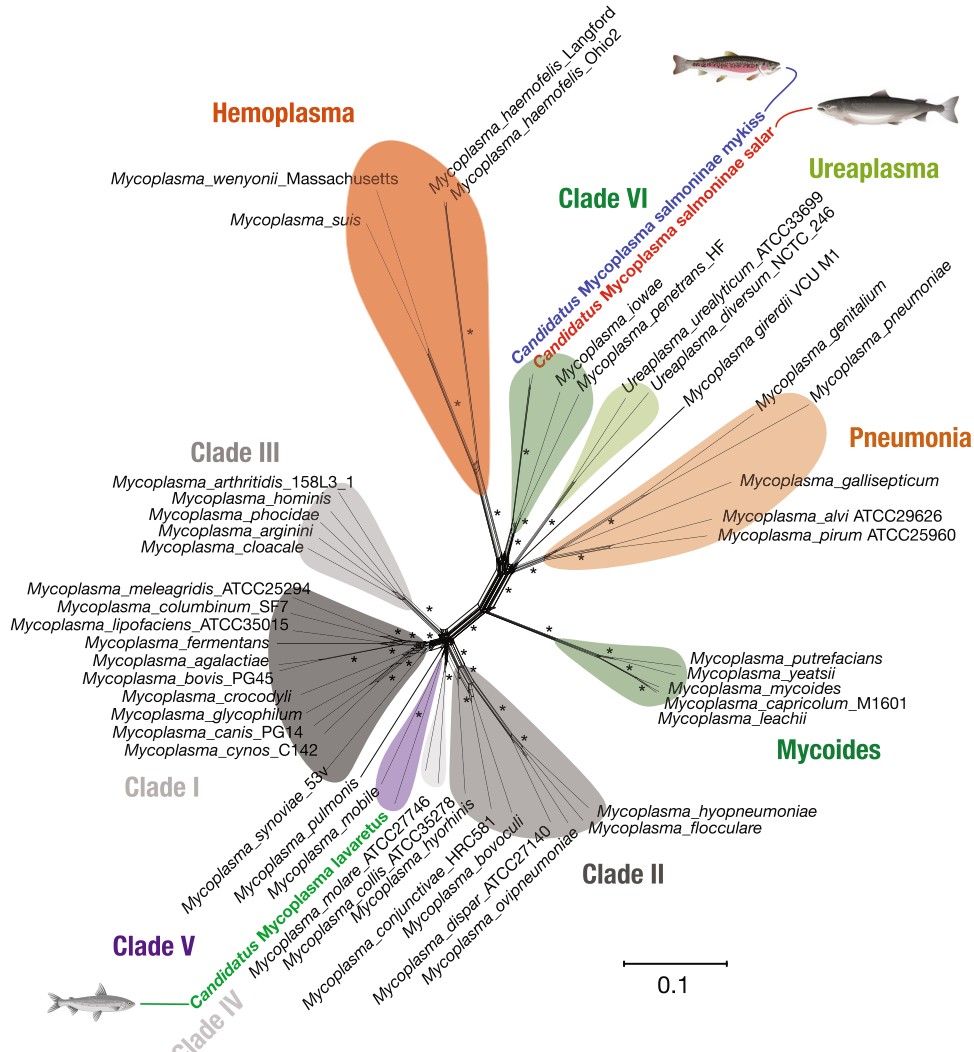

**Fig. 2 Phylogenomics of salmonid *Mycoplasma* MAGs.** Phylogenetic network analyses of the bacterial genomes collected. Asterisks represent bootstrap branch support of 100. Branch lengths are measured in the expected number of substitutions per site. *Bacillus pumilus ATCC7061* was used as an outgroup for all phylogenomic analysis but is hidden for clarity. Blue, red, and green font indicate the salmonid MAGs of *Candidatus* Mycoplasma salmoninae mykiss, *Candidatus* Mycoplasma salmoninae salar, and *Candidatus* Mycoplasma lavaretus, respectively. Suggested clades of phylogeny are distinguished by colouring to increase clarity.

Comparison of ANI, environmental relation, and host group of *Mycoplasma* revealed that *Mycoplasma* are not only clustering according to phylogeny, but also the origin of host and type of tissue, further emphasising niche-specific adaptations (Fig. 3 and Supplementary Fig. 2a).

**Metabolic reconstruction of salmonid MAGs of *Mycoplasma* suggests adaptation to host environment.** Using KEGG we were able to annotate 59.8%, 49.4% and 55.1% of the genes for MSM, MSS and ML, respectively. Especially singletons missed KEGG annotation, indicating that novel functions are yet to be described for many genes in these MAGs (Supplementary Fig. 2a).

Comparison of shared KEGG annotations among MSM, MSS, ML and their nearest relatives *M. mobile* 163 K and *M. iowae* 695 revealed 247 shared KEGG functions among the genomes (Fig. 2 and Supplementary Fig. 3b). Investigation of present KEGG annotations revealed that fermentation of sugars through glycolysis appeared to be the main method of ATP production in MSM, MSS and ML. As in many other *Mycoplasma* species, the genomes are characterised by reduced metabolic functionality as

all of the genomes lack general functions, such as the citric acid cycle. Together, these findings are in line with conserved adaptations to, and dependence on, the host gastrointestinal environments across the salmonid related MAGs and their nearest relatives.

Unravelling functions of salmonid MAGs revealed several putatively beneficial functions for their salmonid hosts, including thiamine ($B_1$) biosynthesis[26], riboflavin ($B_2$) biosynthesis[27] and polyamine metabolism[28]. Interestingly, we found a complete pathway of isoprenoids biosynthesis by the non-mevalonate (MEP) pathways in two of the salmonid MAGs, including MSM and MSS (Supplementary Fig. 6 and Supplementary Fig. 7). The MEP pathways are rarely found in *Mycoplasma*, except for the intestinal associated *M. iowae* and *M. penetrans*, the sister group to MSM and MSS[29,30]. We hypothesise this is to reduce the need to obtain isoprenoid precursors from the host and an adaptation towards intestinal environments[22].

In brief, our genetic findings are in accordance with a model where *Mycoplasma* is functionally adapted to the environment in the gut of salmonids. In all three salmonid MAGs, we found

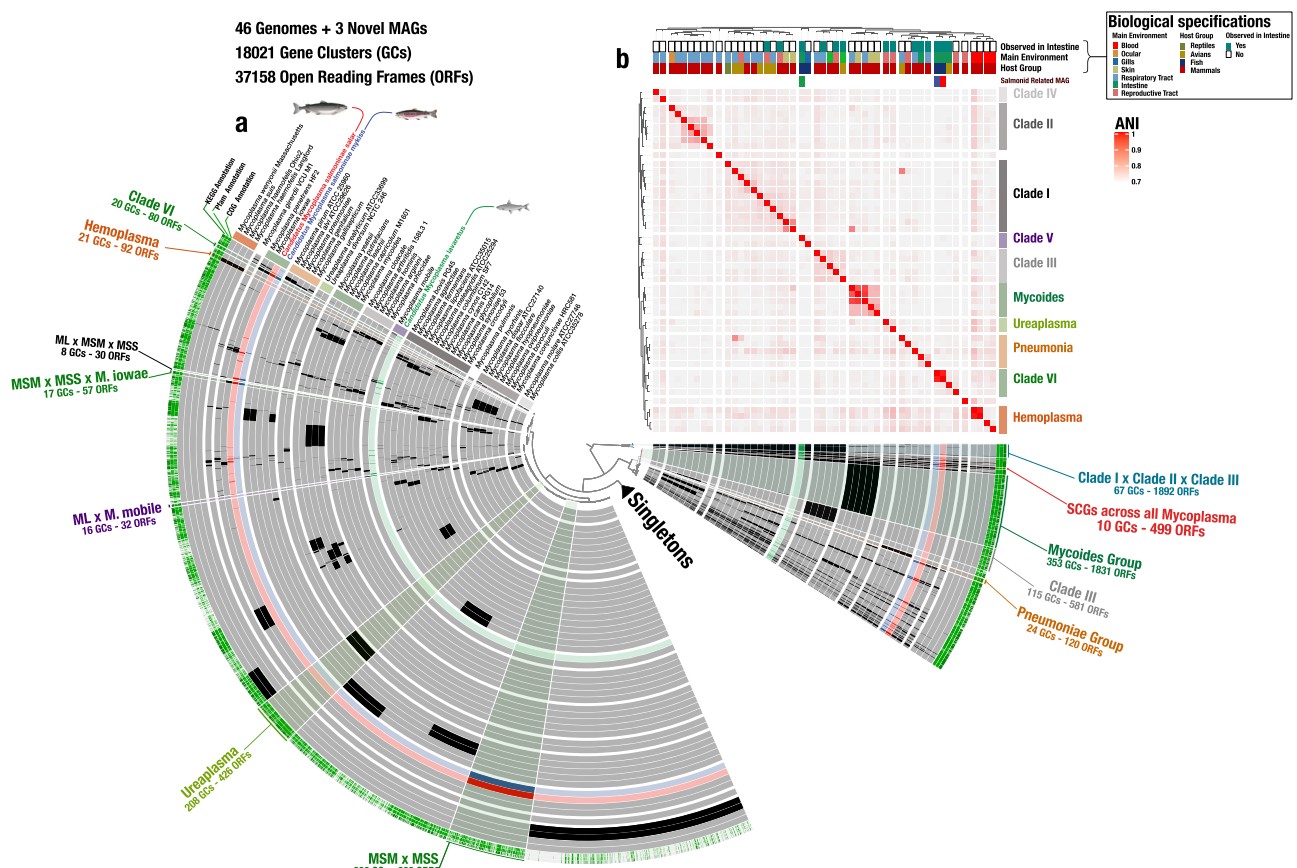

**Fig. 3 Comparative genomics of three salmonid co-assembled MAGs with 49 different *Mycoplasma* genomes. a** Circle diagram based on presence/absence of the 18,021 gene clusters (GCs) containing one or more genes contributed by one or more isolated genomes. Bars in the 49 layers indicate the occurrence of GCs in a given isolated genome and MAGs, black bars indicate presence of GCs, whereas grey indicates absence. Blue, red and green font and layer colour indicate the salmonid MAGs of *Candidatus* Mycoplasma salmoninae mykiss, *Candidatus* Mycoplasma salmoninae salar, and *Candidatus* Mycoplasma lavaretus, respectively. GCs are organised based on their distribution across genomes, and layers of genomes are organised based on their average nucleotide identity (ANI), using Euclidean distance and ward ordination. Selections of co-clustering of genomes were presented in the outer layer and annotations were coloured according to previous clade colours and ANI clusters. Dark red notation indicates single copy core genes (SCGs) between all genomes and MAGs. **b** Heatmap indicates ANI clusters of *Mycoplasma*, where red indicates higher similarity between genomes. Colouring of ANI clusters follows the clade definitions shown in the description in Fig. 2. Biological specifications: The three rows describe the biological designation for each genome, including intestinal relation, main environment and host group.

*uvrABCD*, the global genome nucleotide excision repair system (GG-NER). GG-NER is known to protect bacteria against bile salts in gastrointestinal environments[31]. We found several complete defence systems across the salmonid MAGs (Supplementary Fig. 6), including the stringent response, which is known to react to multiple stress conditions, including amino acid starvation. We found evidence of complete subsystems for lipoic acid metabolism in genomes of clade VI, including MSM, MSS and *M. Iowae* 695. Lipoic acid metabolism is known to be important for oxidative stress response, in agreement with an adaptation against oxidative stress in the gut (Supplementary Fig. 6 and Supplementary Fig. 7). Furthermore, we found presence of the *prtC* gene in all genomes of clade VI, including MSM and MSS, which encodes a putative collagenase, responsible for mucus degradation in *Helicobacter pylori* (Supplementary Data 3). The presence of *prtC* indicates that MSS and MSM are able to live in gastrointestinal environments by facilitating degradation of mucus in the intestine. Interestingly, we found genetic evidence for a cellobiose and chitobiose degrading complex, known as the cellulosome encoded by *celABC*, in ML, MSM and MSS. The closest homologues of *celABC* found in ML, MSM and MSS were found in *M. iowae* 695 with identities ranging from 58.3 to 64.3%. The cellulosome is responsible for

degrading complex polymers, like cellulose, hemicellulose, and chitin[32], indicating that intestinal related salmonid MAGs have some putative ability to degrade long chain polymers in the gut, possibly originating from host mucus layers or host diet (Supplementary Data 3).

All three salmonid MAGs lack oligosaccharide ABC transporters, which are otherwise found in other *Mycoplasma* genomes, indicating that salmonid *Mycoplasmas* are relying on the phosphotransferase system (PTS), like *celABC* (Supplementary Fig. 6 and Supplementary Fig. 7). This suggests that the main sources of energy absorbed by the *Mycoplasmas* from the gastrointestinal tract in its teleost host consist of long-chain polymers, fatty acids, lipoproteins and proteins.

Though the molecular basis of *Mycoplasma* pathogenicity remains largely elusive, we investigated the presence of *Mycoplasma* related pathogenicity factors, including the presence of *traG/traE*[33], *glpF*[34], *katE*[35], *oppA, mgpA/mgpC*[36], virulence factor BrkB, toxins, antitoxins, large membrane proteins (LMPs) and adhesion-related proteins. Our investigation revealed a lack of surveyed putative virulence factors in both MSM and MSS. We found a three-gene cluster with virulence factors, including virulence factor BrkB, an anti-toxin and *glpF*, in ML, indicating

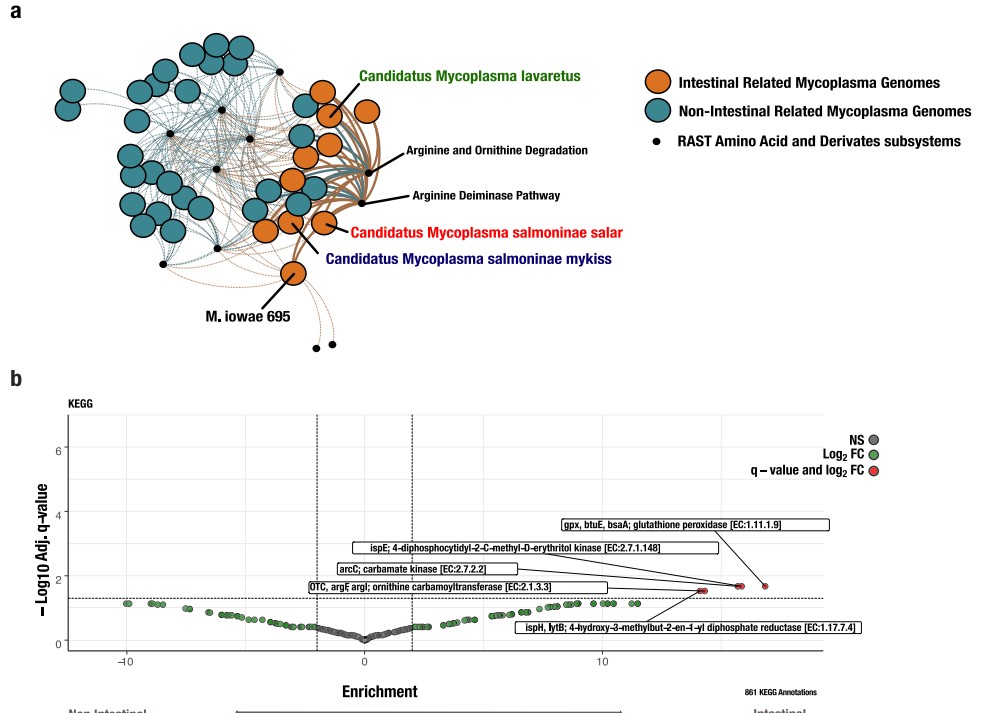

**Fig. 4 Functional enrichment analysis between intestinal related *Mycoplasma* species and non-intestinal related *Mycoplasma* species. a** Genome clustering network based on the RAST 'Amino Acid and derivatives' subsystem. Clustering of genomes was based on RAST 'Amino Acid and derivatives' subsystems present among *Mycoplasma* genomes. The network layout was based on Force Atlas 2 algorithms. Blue colour indicates *Mycoplasma* genomes not being related to intestinal environments, whereas orange nodes indicate *Mycoplasma genomes* being related to intestinal environments. Small black nodes indicate RAST subsystems. Dashed edges indicate non-significant relations. Thick solid edges indicated significance between the RAST subsystem and intestinal environment. Blue, red and green font indicate the salmonid MAGs of *Candidatus* Mycoplasma salmoninae mykiss, *Candidatus* Mycoplasma salmoninae salar, and *Candidatus* Mycoplasma lavaretus, respectively. **b** Volcano plot of 861 KEGG annotated genes in *Mycoplasma*. Investigated genomes were divided into intestinal and non-intestinal related genomes of *Mycoplasma* according to Supplementary Data 2. Functional enrichment was inferred using a generalised linear model (GLMs) with the logit linkage function to compute an enrichment score and *p* value for each function. False detection rate correction to p-values to account for multiple tests was done and only genes with adjusted *q*-values below 0.05 and an enrichment score above 1 are reported.

that ML still possess some level of pathogenic potential, whereas we found no evidence for pathogenicity of MSS and MSM to its host (Supplementary Fig. 8).

**Functional enrichment analysis suggests that intestinal related *Mycoplasma* species are relying on amino acid synthesis, isoprenoid synthesis and an antioxidative protective system.** We performed a functional enrichment analysis by reconstructing metabolic pathways specific for *Mycoplasma* with RAST[37]. This analysis revealed 641, 667 and 676 DNA features, including protein-encoding genes and RNA coding genes, in MSM, MSS, and ML, respectively. Our pathway-based comparison among *Mycoplasma* genomes revealed that *Mycoplasma* species have a broad range of different functionalities (Supplementary Data 2), which fits the high dissimilarity of the phylogenetic and pangenomic analyses and the hypothesised host adaptation (Fig. 3).

Interestingly, we found a significant enrichment of the subsystems corresponding to arginine biosynthesis in *Mycoplasma* species and MAGs associated with intestinal environments including the three MAGs described here (87.5%) compared to those in other environments (27.3%) (Fig. 4a, b). Our enrichment analysis also confirmed a higher prevalence of genes encoding the MEP pathway in intestinal environments (Fig. 4b). Lastly, our analysis revealed that glutathione peroxidase was significantly over-represented in *Mycoplasma* species associated with intestinal environments, indicating that antioxidative

protective systems have a putative defensive role in intestinal related *Mycoplasma* (Fig. 4b) (Supplementary Data 4).

## Discussion

Our study presents *Mycoplasma* MAGs characterised from gastrointestinal samples of 12 individuals from 3 different salmonid species. These salmonid related MAGs systematically represented the dominant taxa of the gut microbiota in the three host species. Moving beyond 16S rRNA gene amplicon surveys, we provide the first description of the potential functional importance of *Mycoplasma* related to the gastrointestinal environment of multiple salmonid host species, including an essential amino acid metabolism of putative advantage to their fish host. We acknowledge that MAG technology comes with limitations, resulting in potential false positives. To minimise these limitations, we used state-of-the-art binning methods, combining CONCOCT and manual curation with anvi'o and applied good practice for the reliable generation of MAGs[38]. Our *Mycoplasma* MAG resolved from Atlantic salmon is largely in accordance with another recent assembly[20]. Together, these findings support a non-neutral evolutionary relationship and warrant further investigations into the use of these *Mycoplasma* candidate species and their potential to boost gut health and growth performance in commercial production of fish and potentially other animal species. Our 16S rRNA gene-based qPCR investigation of bacterial load in rainbow trout indicated low biomass of bacteria, which we hypothesise reflects the young age of rainbow trout, a relatively sterile RAS

environment and that the bacteria present are in a phase of initial colonisation of the gut. The dominance of Mycoplasma in this early life stage further indicates that Mycoplasma is important in young salmonids.

Modifications of the microbiota for a putative gain of beneficial phenotypes, such as increased feed efficiency and host disease resilience, have been exploited for livestock productions. We envisage that our findings will further provide a background for a better understanding of the interaction between *Mycoplasma* and aquaculturally relevant salmonids and other teleost species to increase the efficiency of aquaculture production. In Atlantic salmon, the abundance of *Mycoplasma* has been shown to be strongly decreased by the presence of pathogens, which indicates that the abundance of *Mycoplasma* is positively associated with improved growth[8,39], carotenoid utilisation[1] and disease resilience of its host[40]. Together, this suggests that *Mycoplasmas* may be used as a putative biomarker for monitoring performance and disease status in farmed salmonids[6,40].

The phylogeny of salmonid related MAGs of *Mycoplasma* corresponds to the phylogeny of their hosts, Coregoninae and Salmoninae, indicating putative host-specific adaptations. We deciphered genetic and functional differentiations among *Mycoplasma* species, using state of the art methods for phylogenomics, comparative genomics, and reconstruction of metabolic subsystems. Despite the fact that *Mycoplasma* is often reported to be an obligate parasite, our findings revealed that the salmonid-related *Mycoplasma* species could be specifically adapted for ammonotelic hosts, such as most teleosts, due to the ability to utilise ammonia in the gut. We hypothesise that this might have facilitated an evolutionary beneficial relationship between *Mycoplasma* and its salmonid hosts. Numerous recent studies have revealed a strong dominance of a salmonid *Mycoplasma*[6,8,11,13]; here we add to these intriguing findings by presenting draft genomes, broader phylogenetic relationship and functional potential. A high relative abundance of *Mycoplasma* has been associated with higher health status in Atlantic salmon indicating a potential adaptive advantage of *Mycoplasma* to the host fish[40]. Further, neutral modelling approaches comparing environmental and intestinal frequency distributions of *Mycoplasma* in Atlantic salmon have previously suggested that Atlantic salmon related *Mycoplasmas* are well adapted to colonisation of their hosts[41]. Recent findings of marine *Mycoplasma* species and uncultured Tenericutes have shed new light over the evolutionary processes and pathogenicity of *Mycoplasma*, which is corroborated by our findings of symbiotic species of *Mycoplasma* present in marine environments and marine living vertebrates[22,42].

Previous studies have shown the importance of arginine and its derivatives, citrulline and ornithine, in the gastrointestinal tract of farmed fish[43–45]. Biosynthesis of arginine, citrulline and ornithine has previously been investigated in *Mycoplasma*, since this pathway is an important energy source in other *Mycoplasma* species[22]. We found genetic evidence of MSS, MSM and ML being able to use ammonia as a substrate for biosynthesis of ornithine and citrulline by the presence of the genes encoding carbamate kinase (*arcC*) and ornithine transcarbamylase (*otc*). We hypothesise that this is a beneficial trait for salmonids since they lack the ability to de novo synthesise arginine and in addition, this ability of *Mycoplasma* could increase ammonia detoxification in the gut[46], where ammonia are found excessively in salmonids[47]. Furthermore, ornithine uptake from the gut can lead to increased growth in Atlantic salmon[43]. These findings indicate that the presence of *Mycoplasma* in the gut can boost the metabolism of its salmonid host, which we hypothesise could be a result of ammonia detoxification and could result in the increased upper limit of feed utilisation and thereby increased growth of its salmonid host. Further investigations of gene activity related to

ammonia detoxification, isoprenoid synthesis and polymer degradation by *Mycoplasma* would require meta-transcriptomics and metabolomics to further our functional understanding of the role that *Mycoplasma* plays for its host.

Previous studies investigating the biogeographical dynamics of the intestinal microbiota in Atlantic salmon revealed dominance of *Mycoplasma* in adults and returning adult salmon, despite the low presence of *Mycoplasma* in the early life stage of Atlantic salmon[13]. Combined with our results, we hypothesise that the dominance of *Mycoplasma* in salmonids is a result of holobiont evolution. Indeed, we show that these salmonid *Mycoplasmas* harbour genes able to degrade long-chain polymers, such as chitin, which is often abundant in insects and crustaceans that make up an important proportion of the natural diet of juvenile salmonids[48]. This ability of *Mycoplasma* could be beneficial for its host since the degradation of long-chain polymers boosts the nutritional value of a chitin rich diet and therefore could be a co-evolutionary driver between the salmonid hosts and *Mycoplasma*. This hypothesis may also explain the increase of *Mycoplasma* in aquaculture cohorts, where an increase of *Mycoplasma* was shown in the intestinal region of rainbow trout reared on an insect-based diet[8] and an insect-based diet has subsequently proven beneficial in aquaculture[39]. We note that our findings could represent the foundation for an optimised production that exploits the beneficial functions of *Mycoplasma* presence by targeting not only the essential requirements of the fish host but also the requirements of the salmonid specific *Mycoplasma*.

While several previous studies have demonstrated that *Mycoplasma* species are commonly found in salmonids using gene marker technologies, our study resolves the phylogeny and functionality of three MAGs of two species of *Mycoplasma* related to aquaculturally relevant species. Our study facilitates a deeper understanding of the functionality of *Mycoplasma* in salmonids. Furthermore, our results are in accordance with a mutualistic relationship between *Mycoplasma* and its salmonid host and imply that *Mycoplasmas* likely play an important role in nutrition utilisation and possibly health.

## Methods

**Sample collection and DNA extraction**. Sample collections for rainbow trout were carried out at the Research facility of BioMar in Hirtshals, Denmark. Samples from eight individual Atlantic salmon were obtained from a production site in a fjord near Bergen, Norway. The entire gastrointestinal tract was carefully dissected out with sterile tools and gut content was then sampled and preserved in Zymo DNA/RNA shield (Zymo Research). Extraction of DNA from three individuals, including mid and distal intestinal region, of rainbow trouts was carried out using ZymoBiomics DNA miniPrep (Zymo Research), following the protocol of the manufacturer. Extraction of DNA from Atlantic salmon was carried out using MagAttract® PowerSoil® DNA KF Kit, following the protocol of the manufacturer. Samples from a single individual of European whitefish were taken from Lake Suohpatjavri, Norway. The entire gastrointestinal tract was carefully dissected out with sterile tools and gut content was then sampled and preserved in 90% Ethanol. Extraction of DNA from European whitefish was carried out in technical triplicates, using MagAttract Power Soil DNA Kit (Qiagen) with a modified protocol[49].

**Ethical approval**. The methods for rainbow trout were performed in accordance with relevant guidelines and regulations and approved by The Danish Animal Experiments Inspectorate, under license No. 2015-15-0201-00645. The study is thus approved under the Danish law regarding experimental animals. Atlantic salmon and European whitefish included in this study were sacrificed immediately upon catch with a solid hit to the neck region resulting in instant death before tissue and gut content samples were taken. While no particular license is required for such sampling, we stress that all fish handling was supervised by experienced and trained staff in accordance with normal and legal procedures in Norway. We obtained permission for gill net fishing in Lake Suohpatjavri from the County Governor of Finnmark with legal authority through LOV 1992-05-15 nr 47, 113. Fish were euthanized by means of a cerebral concussion prior to sample collection. No ethical permission is required from the Norwegian Animal Research Authority for collection with gill nets and the associated sacrifice of fish (FOR 1996-01-15 nr 23, the Norwegian Ministry of Agriculture and Food).

**Biomass monitoring of bacteria in rainbow trout, using real-time PCR of bacterial V3–V4 16S rRNA gene**. Real-time PCR (qPCR) was performed in 20 µl reactions containing either 1 µl 1:1, or 1:10 DNA template, and 1× AccuPrime SuperMix II (Invitrogen), 6.5 µl ddH$_2$0, 0.1 µM forward, and reverse primers (341 F: 5′-CCTAYGGGRBGCASCAG-3′, 805R: 5′-GGACTACNNGGGTATC-TAAT-3′), and 1 µl of SYBR Green/ROX solution (Invitrogen). qPCR amplifications were performed on an Mx3005 qPCR machine (Agilent Technologies) with the following cycling conditions: 95 °C for 10 min, followed by 40 cycles of 95 °C for 15 s, 55 °C for 20 s, and 72 °C for 40 min.

**Imaging bacteria in mid- and distal gut sections of rainbow trout**. To visualise bacteria in the fish gut sections, formalin-fixed tissue sections were deparaffinised with xylene, dehydrated with 70 and 96% ethanol, respectively, for 3 min each time, and air dried. Vetcashield (Vector Laboratories), an antifade mounting medium with DAPI was added along with a coverslip. Microscopy and image analysis were with an Axioplan II epifluorescence microscope (Carl Zeiss) equipped for epifluorescence with a 100-W mercury lamp. Confocal images were captured on a Leica SPX-5 (Supplementary Fig. 4).

**DNA sequencing**. Fragmentation of DNA was carried out, using Covaris M220 with microTUBE-50 AFA Fibre Screw-Cap. Samples were normalised prior to library preparation. Library preparation was based on Single-tube library preparation for degraded DNA[50]. Prior to the indexing of libraries, all libraries were analysed with quantitative PCR to estimate optimal cycle settings on a Mx3005P qPCR System (Agilent Technologies). Indexed libraries were quality controlled with a bioanalyzer 2100 (Agilent Technologies) and sequenced 100PE and 150PE on a BGISEQ-500 or MGI 2000 at BGI Europe.

**Genome-resolved metagenomics**. Raw sequence reads were quality controlled, using FastQC/v0.11.8 to assess filtering and quality steps. Saturation of sequencing depth to obtain >5X coverage for all reads were estimated for available data, using khmer[51]. Removal of adapters and low-quality reads was done with Adapter-Removal/v2.2.4, with quality base of 30 and a minimum length of 50 bp. Duplicates were removed, and reads were re-paired to remove singletons, using bbmap/v.38.35. In order to increase assembly efficiency by reducing eukaryotic contaminants, data were filtered for the phiX174 genome, Human (HG19) genome, and the respective host genome (Supplementary Table 2), using minimap2[52], using default parameters for short accurate genomic reads. Filtered data were initially single assembled to investigate individual variations of *Mycoplasma* and were also co-assembled to obtain highest completion of MAGs for comparative and phylogenomic analysis, using MegaHIT/v.1.1.1 with a minimal length of 1000 bp per scaffold, using meta-sensitive flag for metagenomic purpose and assembled contigs were quality controlled with Quast/v.5.02. To increase effective binning, we used the anvi'o pipeline[53] (available from http://merenlab.org/software/anvio). The subsequent workflow was outlined in http://merenlab.org/2016/06/22/anvio-tutorial-v2/. Briefly, (I) anvi'o was used to profile the scaffolds using Prodigal/v2.6.3[54] with default parameters to identify genes and HMMER/v.3.3[55] to identify genes matching archaeal[56], Protista (based on http://merenlab.org/delmont-euk-scgs) and bacterial[56] single-copy core gene collections. Also, ribosomal RNA based HMMs were identified (based on https://github.com/tseemann/barrnap). The HMMs were used to determine the completeness of metagenome-assembled genomes (MAGs); (II) Kaiju[57] was used with NCBI's non-redundant protein database 'nr' to infer the taxonomy of genes (as described in http://merenlab.org/2016/06/18/importing-taxonomy/); (III) we mapped short reads from the metagenomic set to the scaffolds using BWA/v0.7.1596 (minimum identity of 95%) and stored the recruited reads as BAM files using samtools[58]; (IV) anvi'o profiled each BAM file to estimate the coverage and detection statistics of each scaffold, and combined mapping profiles into a merged profile database for each metagenomic set. Contigs were binned automatically, using CONCOCT[59], by constraining the number of clusters per metagenomic set to 10. Each CONCOCT bin was manually curated using the anvi'o interactive interface to ensure high completion and low redundancy. The interface considers the sequence composition, differential coverage, GC-content and taxonomic signal of each scaffold[53,60]. Identification of Mycoplasma MAGs was initially conducted by three approaches: (I) by short-read mapping with Kaiju, giving the origin of each scaffold, (II) if ribosomal 16S gene were present, we blasted these to nearest relative, (III) we used HMM hits of single-copy core genes to identify bacterial taxa. Relative abundance of each MAG was calculated based on percentage read recruitment across all samples from the specific host. Quality assessment of MAGs was carried out with anvi'o and checkM[61].

A summary for each metagenome and *Mycoplasma* MAGs generated for this study is available at https://doi.org/10.6084/m9.figshare.13019507 and https://doi.org/10.6084/m9.figshare.13019477, respectively.

**Phylogenomics of *Mycoplasma***. Metagenomic co-assemblies from rainbow trout, Atlantic salmon and European whitefish were binned using CONCOCT and each MAG was manually curated using anvi'o v6.1. The MAGs annotated as *Mycoplasma* were isolated from the dataset for further analysis. To examine the phylogenetic position of these unknown *Mycoplasma* MAGs, 44 known *Mycoplasma* species from different hosts and tissues were selected. We also included two

samples of the closely related *Ureaplasma diversum*, and a sample of *Bacillus pumilus* for use as an outgroup (Supplementary Data 1). We repeated the extraction of HMM hits using HMMER/v.3.3 from each of these additional samples, resulting in 67 orthologous single-copy core genes, using the default anvi'o bacterial profiles, which were aligned using MACSE/v2[62]. Highly divergent taxa were identified using TreeShrink[63], followed by the second round of alignment of the raw data excluding the flagged taxa. To further minimise the stochastic error due to substitution saturation, we removed codons in which the most common translated amino acid was different for more than half of the taxa[64]. We also filtered codons with >50% missing data and assessed overall substitution saturation using PhyloMAd[65]. The filtering steps resulted in 55 acceptable alignments of core genes.

Phylogenomic analyses were performed using three different models of genome evolution: (I) assuming an identical tree across concatenated genes; (II) assuming free recombination across genes; (III) allowing non-treelike evolution due to recombination, hybridisation or lateral gene transfer.

Phylogenetic analysis was performed on a concatenated data set including the final filtered gene alignments. The substitution models for each gene were selected automatically from the GTR + F + Γ + I + R family of models, allowing gene-models to merge into the best fitting partition scheme[66]. This step of partition selection was followed by maximum likelihood phylogenetic inference using a model of proportional variation in branch lengths across partition subsets[67] implemented in IQ-TREE v1.7[68]. The statistical support for branches was estimated using an Shimodaira–Hasegawa-like approximate likelihood-ratio test (aLRT)[69]. Branch supports were examined further using internode certainty support metrics, which are based on Shannon's entropy and indicate the degree of certainty for each branch considering the two most prevalent branches in the data[70]. Internode certainties for gene trees and for sites were calculated independently, using estimates of site- and gene-concordance factors from IQ-TREE[71].

Further phylogenetic analyses were performed for comparison, focusing on estimates of gene trees as a starting point. Individual gene trees were estimated with the best fitting substitution model of the GTR + F + Γ + I + R family in each case, using IQ-TREE v1.7. Species tree inference was then performed assuming free recombination among genes under the multispecies coalescent model, implemented in ASTRAL[72]. In input trees, all branches with aLRT support below 50 were collapsed, thereby minimising the influence of substantial stochastic error in gene trees arising from having a finite sample of nucleotide sites. Analysis under the multispecies coalescent led to a nearly identical phylogenetic inference to using a concatenated alignment (Supplementary Fig. 2).

Any incongruence in phylogenetic signals across the data was further explored using phylogenetic network analysis. We estimated a network from the concatenated alignment of genes using the Neighbour-Net algorithm[73], with pairwise genetic distances calculated under the F81 substitution model and empirically calculated base frequencies, implemented in SplitsTree v4.16[74]. Statistical supports for network branches were calculated using 1000 bootstrap replicates.

**Comparative genomics of *Mycoplasma***. *Mycoplasma* genomes from different species were selected based on completion level. Only genomes above 80% completion and below 10% redundancy were selected from Genbank. Selected genomes are referred to as external genomes (Supplementary Data 1).

These external genomes were compared with salmonid MAGs, using anvi'o/v6.1 as the previous studies[60]. Similarities of each amino acid sequence in every genome were calculated against every other amino acid sequence across all genomes, using blastp. *minbit heuristics* of 0.5 were implemented to eliminate weak matches between two amino acid sequences[75] and a mcl inflation of 10. The MCL algorithms were used to identify gene clusters in amino acid sequence similarity search results[76]. Euclidean distance and ward linkage were used to organise gene clusters and genomes. ANI was calculated, using PyANI[77]. A summary of the pangenome generated for this study is available at https://doi.org/10.6084/m9.figshare.13019543.

Openness estimates calculations of pangenome were calculated, based on Heaps' law, using R-package micropan[25,78], calculations were carried out with 1000 permutations.

**Functional description of salmonid MAGs**. To increase functional knowledge of the *Mycoplasma* genomes, we annotated functions, using Pfam[79], COG[80], KEGG[81] and RAST[37]. RAST was implemented successfully for 39 of the genomes to minimise *Mycoplasma* specific readthrough of the UGA stop codon[82,83].

Functional enrichment analysis of KEGG annotated genes was carried out on all genomes, using generalised linear models (GLMs) through anvi'o/v6.1. GLMs were carried out with the logit linkage function to compute an enrichment score and p value for each function. False detection rate correction to p-values to account for multiple tests was done using the R package qvalue.

Network analysis of genomes based on amino acid metabolism was based on RAST subsystems. Gephi/v0.9.2[81] was applied to generate a functional network using the Force Atlas 2 algorithm to connect MAGs and RAST functions with 1,000,000 iterations. Only functional nodes related to amino acid metabolism were kept for analysis.

**Statistics and reproducibility**. Statistics were carried using R/v.4.0.3 and Python/v.3.6, to generate results that are reported in the paper and at https://github.com/JacobAgerbo/Comparative_Mycoplasma.

**Reporting summary**. Further information on research design is available in the Nature Research Reporting Summary linked to this article.

## Data availability

The salmonid related *Mycoplasma* MAGs are publicly available at https://doi.org/10.6084/m9.figshare.13019477. Summaries of each metagenome used to generate salmonid related *Mycoplasma* MAGs are publicly available at https://doi.org/10.6084/m9.figshare.13019507. Summary of pangenome used to analyse salmonid related *Mycoplasma* MAGs is publicly available at https://doi.org/10.6084/m9.figshare.13019543. Supplementary Data 2 contains source data for Fig. 4. The raw dataset generated during the current study are available in European Nucleotide Archive (ENA) repository with project accession numbers PRJEB40990. Genome accession codes for *Candidatus* M. salmoninae mykiss (MSM), *Candidatus* M. salmoninae salar (MSS), and *Candidatus* M. lavaretus (ML) are GCA_905477455, GCA_90547759 and GCA_905477555, respectively. All other data are available from the corresponding authors upon reasonable request.

## Code availability

Any code used for the study used to generate results that are reported in the paper and central to its main claims are available at https://github.com/JacobAgerbo/Comparative_Mycoplasma and on Zenodo at https://doi.org/10.5281/zenodo.4629623.

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

## Acknowledgements

The research was funded by The Independent Research Fund Denmark ("HappyFish", grant No. 8022-00005B), GUDP ("Præ-Pro-Fisk", grant No. 34009-17-1218) and the FHF—Norwegian Seafood Research Fund ("HoloFish", grant No. 901436). We would like to thank Anni Nielsen, BioMar Trial coordinator, ATC Hirtshals, Denmark and Torunn Forberg, Biomar RD, Trondheim, Norway for assisting with sampling rainbow trout, Cathrine Kalgraff from Lerøy is thanked for assisting with salmon samples.

## Author contributions

M.T.L. and J.A.R. conceived the study with input from A.M.B., M.T.P.G. and K.K. Sampling was organised and performed by J.A.R., K.R.V., M.T.L., M.T.P.G., M.D.M., L.C.P., H.S. and K.P. J.A.R., L.C.P., K.R.V., L.V.G.J. and A.M.B. carried out laboratory work. J.A.R., T.O.D and D.A.D. performed the computational analysis. J.A.R. wrote the paper with input from all authors.

## Competing interests

The authors declare no competing interests.
