## [Peer Review File · Communications Biology]

Reviewers' Comments:

Reviewer #1:

Remarks to the Author:

This manuscript describes the assembly and pangenomic analysis of several *Mycoplasma* metagenome-assembled genomes associated with two different species of salmonids. The overall analytical strategy is sound, and the findings expand our understanding of *Mycoplasma* biology and function. However, the hypothesis that a mutualistic relationship exists between *Mycoplasma* and the salmonids would benefit from the collection of orthologous supporting evidence.

Specific comments by line:

72 Please provide a citation.

98 This statement is key. While the MAGs do not disprove this hypothesis, nor are they sufficient to prove it.

137 Seems like on the lower end of what has been observed. Why might this be unusually low?

159 This is speculative. Please provide additional data or quantitative analyses demonstrating this association.

198 This is not an appropriate way of quantifying pangenome openness. See, e.g., <https://pubmed.ncbi.nlm.nih.gov/19086349/>

204 Are there enough representatives to really assess niche adaptation? Or is each genome ~unique?

214 What does "strikingly high" mean? How does the number of shared functions compare to any group of more or less related clades?

217 Please elaborate on how KEGG categories were analyzed.

230 This seems like stronger evidence for ecological association than the general discussion of functions and could be highlighted in a figure (4 or S3, perhaps?)

236 Is this unique to the salmonid-associated MAGs? How was representation addressed in these presence/absence analyses? Overrepresentation could occur when selecting for genomes that are closely related. Was a representative selected from each genome cluster? Filtered by ANI?

250 The highly fragmented nature of the MAGs should be mentioned as a confounder here.

269 The use of the word "adapted" here is confusing. Was the antioxidative protective system previously used for something else? How do you know its presence was due to adaptation?

281-286 This is speculative.

305-317 This is quite solid.

Methods questions:

How were MAGs first identified and binned to actually be mycoplasma?

Why were mags combined? How was this done?

Please provide the CONOCT output as a table.

Is there a read cover saturation curve? This would be helpful to interpret potential effects of sequencing depth on MAG contig #/completeness.

How were the 67 orthologous single copy genes chosen?

Figure 3 might benefit from being broken up or have some kind of more clear break down. Consider marking the genomes of interest again on the right hand side for clarity.

Sfig 3 would benefit from having the y axis separated by category. It is difficult to look at all these colors and pull out specific categories.

Reviewer #2:

Remarks to the Author:

The authors discovered richness of Mycoplasma species in three salmon fish species. Using genomics pipelines, they revealed the gene profiles and phylogenetic positions of the Mycoplasma symbionts as well as heterogenicities between different Mycoplasma genomes. The authors argue that the presence of abundant mycoplasmas in the fish gut established a mutualistic relationship. The evidence is however rather weak. The metabolic network should be reconstructed using the genome annotation. Arginine metabolism is regarded by the authors as a nutrient source rather than a critical approach to energy gain. The authors are suggested to read the recent genomics papers regarding Mycoplasma symbionts in different hosts. The synthesis of vitamins in the mycoplasma genomes cannot support the symbiotic benefit to the host. The author did not present the lost genes due to genome reduction supposed to be a result of symbiosis. The proposed mutualistic relationship is progressively established between mycoplasmas and fish host. The gene loss is one of the indicators of the symbiotic level. The authors are suggested to provide FISH pictures to show the position of the mycoplasmas on gut surface or between intestine villi. To understand the transmission of symbionts, we usually need to determine how they are delivered to next generations. With this evidence, we believe the strong mutualistic tie between symbiont and host. Overall the manuscript lacks novelty and contains some mistakes in understanding of the symbiotic Mycoplasma.

Minor comments:

L91. I don't agree with the streamlined genome of mycoplasma. The small genomes are a result of symbiosis that leads to gene loss. The authors obviously misunderstood the origin of the genome reduction in symbionts.

L104-105, In light of the recent reports, I think this is groundless (see Wang et al BMC genomics. 2020).

L106. The authors did not realize there are numerous findings of mutualistic Tenericutes in teleosts including Karen E. Sullam, Mol Ecol. 2012 July ; 21(13) and Lian CA, Front Microbiol. 2020 Jan 10;10:2978.

L137-140. The authors did not show the rest of microbiomes in the guts. It will be interesting to demonstrate the consistency of microbiomes (including non-mycoplasmas) between individuals.

L141. The statement regarding GC content should be careful. Please refer to Wang et al. BMC genomics 2020). This is an indicator of symbiosis level.

L201-203. Number of singletons depends on the identity used in pairwise comparisons of Anvio. With low identity cutoff, the homologs will be combined. The authors should present the cutoff.

L243-247. What is identity to known homologs? Presence of active domains is also important for the proposed functions.

L252, do the authors mean ABC transporters for oligosacchrides? Please mycoplasmas usually use PTS systems to import sugars.

L296-297. This statement is in wrong direction. Mutualism is much more common.

L299. I think ammonium is a production of arginine degradation.

L310-322. I don't agree with the authors' discussion on arginine metabolism. Please refer to Wang et al. 2016. EM. Vol 18. 2646. Arg is a critical energy source of Mycoplasma. Ammonia and CO₂ are end productions of the metabolism. Ammonia is not toxic because it can be feed into amino acids synthesis. Please identify arcD gene in the MAGs.

L397. What is the parameter for the filtration?

Conclusions: it is too long and lacks insights into the present work.

L419. Are  is

L434. Change resulting to resulted.

Suppl Table S3. GC content should be in percentage to show more accurate numbers.

Rebuttal for COMMSBIO-20-3080

Reviewers' comments:

Reviewer #1 (Remarks to the Author):

This manuscript describes the assembly and pangenomic analysis of several *Mycoplasma* metagenome-assembled genomes associated with two different species of salmonids. The overall analytical strategy is sound, and the findings expand our understanding of *Mycoplasma* biology and function. However, the hypothesis that a mutualistic relationship exists between *Mycoplasma* and the salmonids would benefit from the collection of orthologous supporting evidence.

We thank reviewer #1 for the positive feedback. We are pleased to see that the reviewer found our methods reasonable and that the reviewer finds our results of value for the scientific environment. We have collected orthologous supporting evidence by using Fluorescence Microscopy (FM) to show that most bacteria in the gut lumen are in direct contact with the host's intestinal epithelial surface adding to our interpretation of a putative mutualistic relationship. The identity of the bacteria on the FM images could not be established using more specific probes, which is likely due to the low bacterial biomass impeding the generation of solid signals. Hence, here we just show the DAPI based images. However, while we cannot convincingly conclude that the new FM images depict *Mycoplasma* species, they do add orthologous evidence to our results and the fact that *Mycoplasma* species make up >50% of all microbial reads (Fig. 1B) suggests that at least some of the observed bacterial cells belong to *Mycoplasmas*.

Furthermore, we included qPCR results from bacterial load, using 16S V3-V4 rRNA gene profiling, in the juvenile rainbow trout to emphasise the reported low bacterial load present in the intestinal environment of juvenile rainbow trout, which underline the presence of *Mycoplasma* in a sparse microbiota in rainbow trout, which we hypothesise to be a result of young age, fairly sterile RAS environments, and early stage colonisation of bacteria, dominated by *Mycoplasma*, in the intestinal environment.

We further investigated presence or absence of known virulence factors in salmonid related *Mycoplasma* MAGs. Our additional findings further support a non-stochastic relationship between *Mycoplasma* and their salmonid hosts.

While we argue that our results, including the novel orthologous evidence described above, provide evidence of a mutualistic relationship, we agree that our results do not prove this as such, and we have adjusted our wording accordingly.

Specific comments by line:

72 Please provide a citation.

Thanks to the reviewer for pointing out this. Citations have now been provided.

98 This statement is key. While the MAGs do not disprove this hypothesis, nor are they sufficient to prove it.

This statement has been deleted. Other *Mycoplasma* species have been reported as symbionts, but these species are rarely found. We envisage that our findings will still increase the understanding of *Mycoplasma* as a possible symbiont of relevance for the aquaculture industry.

137 Seems like on the lower end of what has been observed. Why might this be unusually low?

We thank the reviewer for pointing out this. As we write this is low, but is similar to other metagenomic assembled *Mycoplasmas*, like <https://www.frontiersin.org/articles/10.3389/fmicb.2019.02978/full>, where the G+C content is 23.5%. Therefore, we correct the reporting of “unusually low”. We do think the low GC content is a result of the symbiotic interaction between *Mycoplasma* and its host, as the other reviewer proposes, see line 138-140.

159 This is speculative. Please provide additional data or quantitative analyses demonstrating this association.

We have collected orthologous supporting evidence by using fluorescence microscopy imaging to localise putative *Mycoplasma* in close contact to the gut epithelia in rainbow trout. Our findings revealed that bacteria, very likely *Mycoplasma*, being in close contact with the host epithelial surface. These findings further support a non-neutral relationship between *Mycoplasma* and its salmonid host, please see lines 145-152.

198 This is not an appropriate way of quantifying pangenome openness. See, e.g., <https://pubmed.ncbi.nlm.nih.gov/19086349/>

We thank the reviewer – this is a great point. We included alpha calculations for gene clusters of the pangenome, using the R-package *micropan* <https://pubmed.ncbi.nlm.nih.gov/25888166/>, which is based on <https://pubmed.ncbi.nlm.nih.gov/19086349/>. These calculations confirmed our findings of an open pangenome, please see line 215. In average, each genome added 368 new gene clusters, indicating an open pangenome. This echoes the sentence from Tettelin et al. 2008 (<https://pubmed.ncbi.nlm.nih.gov/19086349/>): “In other words, analysis of the *S. agalactiae* pan-genome revealed that on average each additional genome sequence would reveal 33 previously uncharacterized genes, implying an unbounded or open pan-genome”.

204 Are there enough representatives to really assess niche adaptation? Or is each genome ~unique?

This is a great point. According to our ANI analysis, the genomes are rather unique. Our additional analysis of individual based comparison of *Mycoplasma* genomes between salmonid hosts, where several individuals for Atlantic salmon and rainbow trout were included (Supplementary Fig. 3a), indicate the same ANI pattern as Fig. 3, which we do think support such a niche adaptation.

214 What does “strikingly high” mean? How does the number of shared functions compare to any group of more or less related clades?

This wording has been removed in order to avoid confusion.

217 Please elaborate on how KEGG categories were analyzed.

Presence of KEGG annotation were analysed across the three salmonid related MAGs, which were compared to nearest relatives.

This has been reworded to improve clarity. See line 226.

230 This seems like stronger evidence for ecological association than the general discussion of functions and could be highlighted in a figure (4 or S3, perhaps?)

We agree with the reviewer and we have reworded this and tried to tone down the conclusive wording. Please see lines 244-245. Ecological association to intestinal environment are noted in figure 3, where all intestinal observed *Mycoplasma* is marked under ANI heatmap with a turquoise square.

236 Is this unique to the salmonid-associated MAGs? How was representation addressed in these presence/absence analyses? Overrepresentation could occur when selecting for genomes that are closely related. Was a representative selected from each genome cluster? Filtered by ANI?

This is not specific to salmonid related MAGs, but we emphasise this is important for *Mycoplasma* in salmonids to live in the host. For the analysis, we analysed presence/absence of RAST and KEGG annotations for all genomes. This is not shown in supplementary figure, since we focused the figure on the intestinal related *Mycoplasma*. This has been reworded to improve clarity. Please see Lines 250-251

250 The highly fragmented nature of the MAGs should be mentioned as a confounder here.

We have rechecked completeness and contamination/redundancy of the MAGs with other tools than anvio, including checkM (<https://pubmed.ncbi.nlm.nih.gov/25977477/>), since this tool seems rather conventional and has often been used in other studies, like: <https://www.nature.com/articles/s41564-018-0176-9#Sec11>.

Our new results show that completeness for 2 of 3 MAGs (ML and MSM) were above 95% and below 5% contamination. The MSS MAG had a completeness of >85% and below 5% contamination, which is mostly classified as a high quality MAG; <https://ena-docs.readthedocs.io/en/latest/faq/metagenomes.html>. We have included these new results in table 1.

In the light of these results, we argue that the originally reported values of completeness reflected the method used, and not that the MAGs are highly fragmented.

269 The use of the word “adapted” here is confusing. Was the antioxidative protective system previously used for something else? How do you know its presence was due to adaptation?

Thanks for pointing this out. This has been reworded for clarification. See lines 251-252.

281-286 This is speculative.

We have re-worded this paragraph to highlight the relevance and potential of understanding the genetic functions of key gut microbes in farmed fish too. In the revised version we now consider supporting evidence from previous studies that also show a positive correlation between *Mycoplasma* abundance and fish performance (See Refs below), and then discuss how our genomic analyses add functional knowledge to these previous studies as well.

Rimoldi et al. 2019; <https://www.mdpi.com/2076-2615/9/4/143>

Rimoldi et al. 2021; <https://link.springer.com/article/10.1007/s10695-020-00918-1>

Brown et al. 2019;

<https://www.sciencedirect.com/science/article/abs/pii/S1050464818307988?via%3Dihub>

305-317 This is quite solid.

Thanks, we appreciate this.

Methods questions:

How were MAGs first identified and binned to actually be mycoplasma?

Thanks to reviewer for pointing out this. This is now stated more clearly in the methods section. MAGs were first identified, using CONCOCT and further manually curated using Anvi'o.

Identification of *Mycoplasma* MAGs were initially conducted by three approaches; 1) by short read mapping with Kaiju, giving an origin of each scaffold, 2) If Ribosomal 16S gene were present, we blasted these to nearest relative, 3) We used HMM hits of single copy core genes to identify bacterial taxa. Furthermore, we confirmed the findings of Kaiju, using the KEGG taxonomy of genes in the MAGs. This has now been clarified in lines 455-459.

Why were mags combined? How was this done?

This might be a misunderstanding from our writing, which we have tried to clarify. MAGs were never combined after being identified. We initially ran a single assembly of single samples to identify individual variations of *Mycoplasma* MAGs across the three host types, since we observed that the ANI grouping followed host taxonomy (Fig. S3), we made co-assembly of all samples, independently for each host (e.g., co-assembly of salmon samples, co-assembly of rainbow trout samples, co-assembly of whitefish samples). This way we could use most data available to obtain the best completeness and representatives for each host species. We have clarified this on lines 434-438 in the revised version.

Please provide the CONOCT output as a table.

These tables are provided as a summary of metagenomes in our figshare repository: <https://figshare.com/s/a93fb5a38e6cef40961d>, as cited in lines 550-551 of our revised manuscript.

Is there a read cover saturation curve? This would be helpful to interpret potential effects of sequencing depth on MAG contig #/completeness.

We thank the reviewer for pointing out this. Saturation curves were calculated, using khmer. We estimated the minimum sequencing effort to obtain 5X coverage of all novel reads. We did the calculation, using all reads available per sample instead of random subsample. This information has been added to methods (please see lines 121-122) and supplementary Fig. S1.

<https://khmer-recipes.readthedocs.io/en/latest/004-estimate-sequencing-saturation/>
<https://github.com/dib-lab/khmer/blob/master/sandbox/saturate-by-median.py>

How were the 67 orthologous single copy genes chosen?

We thank the reviewer for pointing out this. We have tried to clarify this better in the method text. The selection and identification of the 67 orthologous single copy genes were carried out, using the default HMM profiles *anvi'o* uses for bacterial and archaeal single-copy core genes with HMMER, as described in <https://merenlab.org/2017/06/07/phylogenomics/pangenomic--phylogenomics>. We chose to initially keep all the single copy genes found by HMMER and filter according to codons in which the most common translated amino acid was different for more than half of the taxa, and filtered codons with >50% missing data to minimise inflation, resulting in 55 genes and alignments of these genes. Please see lines 469-477.

Figure 3 might benefit from being broken up or have some kind of more clear break down. Consider marking the genomes of interest again on the right hand side for clarity.

We do agree with the reviewer that this figure is complex, and we have tried to increase clarity by adding marks of genome of interest. We further marked the layer of ML, MSS, and MSM, green, red, and blue respectively to focus more on presence of gene clusters in ML,

MSS, and MSM. We further coloured the annotations of gene clusters according to the phylogenomic determined clades, which also corresponds ANI annotations.

Sfig 3 would benefit from having the y axis separated by category. It is difficult to look at all these colors and pull out specific categories.

We have reordered the y axis for clarity and concatenated subsystems of tRNA aminoacylation for all amino acids for clarity, please see new figure Sfig 6.

Reviewer #2 (Remarks to the Author):

The authors discovered richness of *Mycoplasma* species in three salmon fish species. Using genomics pipelines, they revealed the gene profiles and phylogenetic positions of the *Mycoplasma* symbionts as well as heterogenicities between different *Mycoplasma* genomes. The authors argue that the presence of abundant mycoplasmas in the fish gut established a mutualistic relationship. The evidence is however rather weak. The metabolic network should be reconstructed using the genome annotation. Arginine metabolism is regarded by the authors as a nutrient source rather than a critical approach to energy gain.

The authors are suggested to read the recent genomics papers regarding *Mycoplasma* symbionts in different hosts. The synthesis of vitamins in the mycoplasma genomes cannot support the symbiotic benefit to the host. The author did not present the lost genes due to genome reduction supposed to be a result of symbiosis. The proposed mutualistic relationship is progressively established between mycoplasmas and fish host. The gene loss is one of the indicators of the symbiotic level. The authors are suggested to provide FISH pictures to show the position of the mycoplasmas on gut surface or between intestine villi. To understand the transmission of symbionts, we usually need to determine how they are delivered to next generations. With this evidence, we believe the strong mutualistic tie between symbiont and host. Overall the manuscript lacks novelty and contains some mistakes in understanding of the symbiotic *Mycoplasma*.

We thank Reviewer #2 for the scrupulous review of our manuscript with focus and expertise on *Mycoplasma*. We have gone through each of the reviewer's comments and suggestions to improve the clarity and quality of our manuscript accordingly. Here, we address each comment and explain in detail how we have responded by adding more results to further improve the manuscript and emphasise its novelty.

We have tried to provide further information to reveal direct contact between *Mycoplasma* and its host and we have collected orthologous supporting evidence by using fluorescence microscopy to localise bacteria, which is most likely *Mycoplasma* in close contact to the host, which support the hypothesised relationship between *Mycoplasma* and its salmonid host, as proposed by the reviewer.

Furthermore, we included qPCR results from bacterial load, using 16S V3-V4 rRNA gene profiling, in the juvenile rainbow trout to emphasis low bacterial load present in the intestinal environment of juvenile rainbow trout, which underline the presence of *Mycoplasma* in a sparse microbiota in rainbow trout, which we hypothesise to be a result of young age, fairly sterile RAS environments, and early stage colonisation of bacteria, dominated by *Mycoplasma*, in the intestinal environment.

We further investigated presence or absence of known virulence factors in salmonid related *Mycoplasma* MAGs. Our additional findings further support a non-stochastic relationship between *Mycoplasma* and their salmonid hosts.

Minor comments:

L91. I don't agree with the streamlined genome of mycoplasma. The small genomes are a result of symbiosis that leads to gene loss. The authors obviously misunderstood the origin of the genome reduction in symbionts.

This is simply a misunderstanding of wording, for which we apologize. We agree with the reviewer and do share the same opinion of *Mycoplasma* evolution. In this case we meant "streamlined genomes" according to being streamlined/adapted/optimised according to the gut environment in which the *Mycoplasma* occurs, which is leading to a gene loss. We changed the wording to minimise confusion of this point.

L104-105, In light of the recent reports, I think this is groundless (see Wang et al BMC genomics. 2020).

We would like to thank the reviewer for providing the literature. We do find the Wang et al. 2020 report highly relevant for our study, but not comprehensive enough to explain the interaction between salmonid specific *Mycoplasma* and salmonids. While we agree that our genomes are not the first high quality *Mycoplasma* genomes, they are the first genome-wide characterisation of the salmonid-associated *Mycoplasma* species, which have been frequently reported in 16S amplicon-based studies. Thus, we find our report highly relevant to minimise the knowledge-gap of the novel *Mycoplasma* species associated with salmonids, which represents specific relevance for the aquaculture field in general. Therefore, we believe that our study will still represent a pivotal landmark for the associated field of aquaculture microbiome research.

L106. The authors did not realize there are numerous findings of mutualistic Tenericutes in teleosts including Karen E. Sullam, Mol Ecol. 2012 July ; 21(13) and Lian CA, Front Microbiol. 2020 Jan 10;10:2978.

We appreciate the references to this relevant literature, which we have included in our revised version. We do though underline that we are mainly investigating the relationship between *Mycoplasma* and salmonids in the wake of the numerous previous studies that have identified positive correlations with *Mycoplasma* abundance and performance of aquaculturally important salmonids (see lines 75-83). We find Lian et al. 2020 very relevant and we have now discussed our own results in the context of their findings, please see line 327-330. We find Sullam et al. 2012 of less relevance to our study, since it only covers the phylum of *Mycoplasma* (Tenericutes), which is only mentioned once in the article.

L137-140. The authors did not show the rest of microbiomes in the guts. It will be interesting to demonstrate the consistency of microbiomes (including non-mycoplasmas) between individuals.

We agree with the reviewer, that this is super interesting. Due to the scope of this study as a follow up on previous *Mycoplasma* related findings, we mainly focus on *Mycoplasma*. Indeed, this was also by far the most dominant bacteria (Fig. 1b) and we think it is a more concise and important story to understand the interactions between salmonids and *Mycoplasma*. The remaining non-*Mycoplasma* reads will be reported in a future manuscript currently under preparation.

L141. The statement regarding GC content should be careful. Please refer to Wang et al. BMC genomics 2020). This is an indicator of symbiosis level.

We thank the reviewer for pointing us towards this relevant study. We corrected the statement about GC content according to the provided report, please see lines 139-144.

L201-203. Number of singletons depends on the identity used in pairwise comparisons of Anvivo. With low identity cutoff, the homologs will be combined. The authors should present the cutoff.

We agree with the reviewer and we do apologise for the lack of reporting. We have now included parameters for this in the method section, please see lines 512-515.

Our analysis relies on a default minbit value of 0.5, by the ITEP implemented function of *anvivo* and not identity between amino acid sequences, since percent identity is not a predictor of a good match, as it does not communicate many other important factors such as the alignment length between the two sequences and its proportion to the entire length of those involved.

<https://bmcgenomics.biomedcentral.com/articles/10.1186/1471-2164-15-8>

We used an inflation cut-off for MCL inflation of “10” because we are working with species within the same genus. To be sure our cut-off did not impact results of the study, you also ran two other pangenomes with an MCL inflation of “1” and “2”, which can be more adequate for comparing distantly related genomes within same family, (<https://merenlab.org/2016/11/08/pangenomics-v2/#running-a-pangenome-analysis>).

We did not find any significant differences in number of single copy core genes and singletons between pangenomes of different MCL inflation cut-offs 1,2, and 10.

L243-247. What is identity to known homologs? Presence of active domains is also important for the proposed functions.

We thank the reviewer for pointing out this. We investigated the homologs of *ceIABC* in ML, MSS, and MSM, using BLASTp. We further investigated active sites, using information of homologues on Uniprot. ML, MSS, and MSM had all the closest protein homology to *Mycoplasma iowae 695*, please see lines 258-259.

Furthermore, to improve transparency we provided all amino acid sequences from gene calls in gene clusters in the supplementary material as an excel sheet, see Supplementary Table 6.

L252, do the authors mean ABC transporters for oligosacchrides? Please mycoplasmas usually use PTS systems to import sugars.

We thank the reviewer for pointing out this. We agree with the reviewer and have corrected this to minimise confusion. Please see lines 264-266.

L296-297. This statement is in wrong direction. Mutualism is much more common.

We appreciate the reviewer's insight on mutualistic relationships. We assume the reviewer refers to our original statement in lines 291-292: "*Mutualistic relationships between Mycoplasma species and their hosts have rarely been reported*". We have now deleted this statement following the reviewer's comment to avoid misunderstandings.

L299. I think ammonium is a production of arginine degradation.

We have changed the discussion to present several possibilities of arginine metabolism and include Wang et al. 2016 to give a better picture.

L310-322. I don't agree with the authors' discussion on arginine metabolism. Please refer to Wang et al. 2016. EM. Vol 18. 2646. Arg is a critical energy source of Mycoplasma. Ammonia and CO₂ are end productions of the metabolism. Ammonia is not toxic because it can be feed into amino acids synthesis. Please identify arcD gene in the MAGs.

Thanks to the reviewer for including relevant literature. We have now discussed these recent findings; <https://pubmed.ncbi.nlm.nih.gov/27312602/> in more detail, please see lines 338-345.

We do agree with the reviewer and previous literature about arginine being an important energy source in some *Mycoplasma* spp. and that ammonia and CO₂ can be the end product of the arginine deiminase pathway and this aspect is now included in our discussion (see lines xxx).

Though, we do hypothesise that part of this pathway (ornithine cycle) can be opportunistic between anabolic and catabolic reactions, according to environment and if ammonia content in the gut lumen are stressfully high, which is not uncommon in farmed fish, that are often fed ad libitum. Then, citrulline could be produced from NH₃ and CO₂, by:

- 1) NH₃ + CO₂ <-> carbamoyl phosphate, by Carbamate kinase (arcC)
- 2) carbamoyl phosphate + L-Ornithine <-> Citrulline + phosphate, by ornithine transcarbamylase (OTC)

Citrulline have shown to be beneficial for rainbow trout according to growth and immunity <https://www.sciencedirect.com/science/article/abs/pii/S0044848619324366>

Since arginine is an essential amino acid for salmonids, it would be a fair rationale that salmonid related *Mycoplasma* have adapted a less arginine-required lifestyle in the salmonid intestine than what the reviewer proposes.

<http://www.fao.org/fishery/affris/species-profiles/atlantic-salmon/nutritional-requirements/en/>

Furthermore, we do not believe these interpretations are necessarily mutually exclusive, and we therefor retain our statement of ammonia being potentially stressful in ammonotelic teleosts, like salmonids, under farmed conditions, and that intestinal bacteria, such as *Mycoplasma*, may be relevant to help the teleost host cope with such stressful concentrations of ammonia.

<https://link.springer.com/article/10.1007%2Fs00360-013-0781-0>

<https://jeb.biologists.org/content/222/24/jeb209882>

https://www.jstage.jst.go.jp/article/bifidus1982/6/1/6_1_15/_pdf

We have not been able to find the antiporter for ornithine/arginine (arcD) in the *Mycoplasma* MAGs, which could indicate that the salmonid related *Mycoplasma* rely on other energy sources e.g. fermentation of complex polymers like chitin from natural feed or mucus layers in the intestinal lumen.

L397. What is the parameter for the filtration?

We thank the reviewer for pointing out this. We used short parameters for short accurate genomic reads. This information has now been added on lines 431-432.

Conclusions: it is too long and lacks insights into the present work.

The conclusions have been reduced in length and included more recent work.

L419. Are  is

Thanks to the reviewer for pointing out this. This has been corrected

L434. Change resulting to resulted.

Thanks to the reviewer for pointing out this. This has been corrected

Suppl Table S3. GC content should be in percentage to show more accurate numbers.

We thank the reviewer for pointing out this. This has been corrected

Referee expertise:

Referee #1: pangenomics and bacterial adaptation

Referee #2: marine microbial evolution and genomics

Reviewers' comments:

Reviewer #1 (Remarks to the Author):

This revised manuscript is significantly improved over the previous version. In particular, the additional microscopy strengthen the argument that there exists a mutualistic relationship between *Mycoplasma* and the salmonids. As the authors acknowledge, this relationship is nevertheless still hypothetical, and the revised wording accurately reflects this.

While many issues have been addressed, some limitations still need to be more fully and explicitly acknowledged. The MAGs were generated using co-assembly of multiple samples, which can yield yield genomes that do not in fact exist (see, e.g., <https://aem.asm.org/content/87/6/e02593-20>). This is irrespective of the completeness and contamination metrics provided. Future studies will be needed to confirm the accuracy of these MAGs. Further, fluorescence microscopy was performed on the distal gut only. Please discuss why these samples and no other sample types were used for the microscopy experiments. Please also further discuss why *Mycoplasma* specifically could not be identified, particularly in the context of limits of detection for this type of visualization.

Reviewer #2 (Remarks to the Author):

This manuscript has been revised by authors to greatly improve it. I am happy to see DAPI picture to show the presence of abundant bacteria within intestine surface. If the author can follow this article to show the FISH result of the mycoplasma, the evidence for the beneficial symbiosis will be more solid <https://aem.asm.org/content/70/10/6166>. Regarding the phylogeny of the MAGs, I suggest the authors to further tone down their novelty since all the clades in Figure 3 were known. Clades I, II, III and IV are affiliated with hominis group and clade VI is within suis group. Throughout the manuscript (including figures), the authors should correct the word font of mycoplasma taxonomic names. For example, use "Candidatus [*italic*] *Mycoplasma salmoninae* [regular] mykiss [regular]" and *Mycoplasma iowae* [*italic*] 695 [regular]. Strain names should never be italic.

Some minor comments:

Line 60. Phylogenetic placement is very clear at the time even without genomes.

Line 89. Here must be specified to pathogenic or symbiotic mycoplasmas as most of environmental mycoplasma genomes were between 1-2 Mbp.

Suppl Figs 1 and 2. I have to complain the tiny words on axes.

Lines 139-141. Please move to discussion.

Lines 174-175. First I suggest to move the sentence to discussion. All symbionts come from environments. There are about 0.1% of Tenericutes in marine waters. This is enough for acquisition from environment unless the authors show evidence of vertical transmission. I think Fig S5a can be a tree for 16S genes that include the closest relatives from fish or environment if the authors want to show the source of the symbiotic mycoplasma.

Lines 191-203. There is an existing nomenclature for mycoplasma groups. Please refer to Wang et al BMC genomics and Oshima J Mol Evol 2007 65: 714. All the genomes in the Fig 3 have been included in different known groups.

Line 263. Do they lack amino acid synthesis genes? This is a remarkable signal of host dependence.

Lines 126, 454, 458, 460, 523,529. Were or are should be was or is! Please check carefully all

other grammar problems.

Line 465. Grammar error, rephrase please. A genome cannot contain a bacterium.

Line 473. Put a reference for substitution saturation.

Line 512. minbit heuristic of 0.5 was implemented is better?

Line 528. Modify qvalue to q-value.

Figure 3 contains too many genomes and results. I am afraid the words are too tiny to be visualized. I suggest the authors to keep only useful information. ANI color bar is missing.

Rebuttal for COMMSBIO-20-3080A

Reviewer #1 (Remarks to the Author):

This revised manuscript is significantly improved over the previous version. In particular, the additional microscopy strengthen the argument that there exists a mutualistic relationship between Mycoplasma and the salmonids. As the authors acknowledge, this relationship is nevertheless still hypothetical, and the revised wording accurately reflects this.

While many issues have been addressed, some limitations still need to be more fully and explicitly acknowledged. The MAGs were generated using co-assembly of multiple samples, which can yield yield genomes that do not in fact exist (see, e.g. <https://aem.asm.org/content/87/6/e02593-20>). This is irrespective of the completeness and contamination metrics provided. Future studies will be needed to confirm the accuracy of these MAGs. Further, fluorescence microscopy was performed on the distal gut only. Please discuss why these samples and no other sample types were used for the microscopy experiments. Please also further discuss why Mycoplasma specifically could not be identified, particularly in the context of limits of detection for this type of visualization.

Answers to Reviewer #1

=> We thank reviewer #1 for the positive feedback. We are pleased to see that the reviewer found our manuscript improved and that the reviewer finds our results of value for the scientific environment.

Reviewer #1: While many issues have been addressed, some limitations still need to be more fully and explicitly acknowledged. The MAGs were generated using co-assembly of multiple samples, which can yield yield genomes that do not in fact exist (see, e.g., <https://aem.asm.org/content/87/6/e02593-20>). This is irrespective of the completeness and contamination metrics provided. Future studies will be needed to confirm the accuracy of these MAGs.

=> Thanks to the reviewer for pointing out this. We do acknowledge that this is an issue using MAGs and an important note, especially without curation of MAGs. Some of these issues will be difficult to overcome without culturing and isolation of the bacteria, which we find beyond the scope of the current investigation.

We agree with the reviewer that MAGs do not necessarily represent “real” genomes found in a cell. MAGs are consensus sequences from an assembly. If there is variability within the targeted population (which occurs in most cases in the environment), then the MAG might not represent any “real” genome, and this regardless of single assembly versus co-assemblies. That being said, it is true that co-assemblies can increase the consensus nature of MAGs, representing the centroid of a population in the sequence space spanning multiple samples instead of a single one.

In the case of our present study, we used state-of-the-art binning with anvio. First of all, MAGs were generated by applying both single assemblies for each sample or co-assemblies

(see supp figure 3). Then, we followed a guideline compatible with what Meziti et al. see as relevant, including: 1) using SCG estimates of genome completeness, confirmed both by anvi'o and CheckM, 2) manual inspection and, if necessary, curation of MAGs using the anvi'o interactive interface (takes into account sequence composition, coverage and taxonomy), 3) We only binned >1,000 bp contigs using CONCOCT and with *a priori* testing using MetaBat2 and BinSanity, 4) We compared MAGs to nearest relative and obtained taxonomic profiles from contigs, using Kaiju and KEGG.

We have included these details and perspectives in the revised discussion (**lines 357-360** in 'COMMSBIO-20-3080B_MS w track-changes.docx').

Reviewer #1: *Further, fluorescence microscopy was performed on the distal gut only. Please discuss why these samples and no other sample types were used for the microscopy experiments. Please also further discuss why Mycoplasma specifically could not be identified, particularly in the context of limits of detection for this type of visualization.*

=> Only distal gut samples were included as these were the only specimens that were suitable for the protocol used. We did in fact attempt to visualize the Mycoplasma spp. using a Fluorescence In Situ Hybridization protocol on formalin fixed gut samples. We employed three probes: EUB338, non-EUB338 and a Mycoplasma-specific probe (Myc). The FISH protocol did not result in a convincing signal by any of the attempts. We believe the following reasons could, at least partly explain why: **1)** The salmonid-associated Mycoplasma only have two 16S rRNA operons, based on the sequence analysis, which will limit the number of target molecules in each bacterial cell. Combined with the relatively low metabolic rate, due to the low temperature, this will limit the overall target available for the probes. **2)** The Myc-probe sequence has not been selected based on an optimized FISH protocol. To do so, a more elaborate in vitro study including a positive control based on a pure culture of the salmonid-associated *Mycoplasma* would be highly useful, yet the salmonid-associated Mycoplasma has not been cultivated in vitro as of now, which complicates the validation of the protocol. Along the same line it would be desirable to include a range of non-salmonid-associated Mycoplasma species to ensure the specificity of the specific probe. All in all, to get the FISH protocol to work convincingly more validation steps should be included yet we find these beyond the scope of the current investigation. We have added more details following these arguments in the revised version (**see lines 173-176** in 'COMMSBIO-20-3080B_MS w track-changes.docx').

Reviewer #2 (Remarks to the Author):

This manuscript has been revised by authors to greatly improve it. I am happy to see DAPI picture to show the presence of abundant bacteria within intestine surface. If the author can follow this article to show the FISH result of the mycoplasma, the evidence for the beneficial symbiosis will be more solid <https://aem.asm.org/content/70/10/6166>. Regarding the phylogeny of the MAGs, I suggest the authors to further tone down their novelty since all the clades in Figure 3 were known. Clades I, II, III and IV are affiliated with hominis group and clade VI is within suis group. Throughout the manuscript (including figures), the authors should correct the word font of mycoplasma taxonomic names. For example, use “*Candidatus* [italic] *Mycoplasma salmoninae* [regular] mykiss [regular]” and *Mycoplasma iowae* [italic] 695 [regular]. Strain names should never be italic.

Some minor comments:

Line 60. Phylogenetic placement is very clear at the time even without genomes.

Line 89. Here must be specified to pathogenic or symbiotic mycoplasmas as most of environmental mycoplasma genomes were between 1-2 Mbp.

Suppl Figs 1 and 2. I have to complain the tiny words on axes.

Lines 139-141. Please move to discussion.

Lines 174-175. First I suggest to move the sentence to discussion. All symbionts come from environments. There are about 0.1% of Tenericutes in marine waters. This is enough for acquisition from environment unless the authors show evidence of vertical transmission. I think Fig S5a can be a tree for 16S genes that include the closest relatives from fish or environment if the authors want to show the source of the symbiotic mycoplasma.

Lines 191-203. There is an existing nomenclature for mycoplasma groups. Please refer to Wang et al BMC genomics and Oshima J Mol Evol 2007 65: 714. All the genomes in the Fig 3 have been included in different known groups.

Line 263. Do they lack amino acid synthesis genes? This is a remarkable signal of host dependence.

Lines 126, 454, 458, 460, 523, 529. Were or are should be was or is! Please check carefully all other grammar problems.

Line 465. Grammar error, rephrase please. A genome cannot contain a bacterium.

Line 473. Put a reference for substitution saturation.

Line 512. minbit heuristic of 0.5 was implemented is better?

Line 528. Modify qvalue to q-value.

Figure 3 contains too many genomes and results. I am afraid the words are too tiny to be visualized. I suggest the authors to keep only useful information. ANI color bar is missing.

Answers to Reviewer #2

=> We thank reviewer #2 for the positive feedback. We are pleased to see that the reviewer found our manuscript improved and finds our results of value for the scientific environment. We thank reviewer #2 for scrutinizing feedback, which we find useful to improve our manuscript.

Reviewer #2: *If the author can follow this article to show the FISH result of the mycoplasma, the evidence for the beneficial symbiosis will be more solid*

<https://aem.asm.org/content/70/10/6166>.

=> We agree with the reviewer, and we appreciate the original suggestions to add this layer of data, also appreciated by both reviewers. Please see our response to reviewer 1 above regarding interpretation of our current FISH results. Lastly, after the reviewer pointed us in this direction in the original reviews, we have indeed established a relationship with other colleagues towards the establishment of pure cultures of these salmonid *Mycoplasma* species based on cell cultures from the host species' gut epithelial tissue.

Reviewer #2: *I suggest the authors to further tone down their novelty since all the clades in Figure 3 were known. Clades I, II, III and IV are affiliated with hominis group and clade VI is within suis group.*

=> Thanks to the reviewer for pointing out this. We have toned down the claim of novelty, as the overall phylogeny of *Mycoplasma* is not novel. However, we do emphasise that species of *Mycoplasma* have been highly under reported in marine hosts, compared to terrestrial hosts (see lines **218-219** in 'COMMSBIO-20-3080B_MS w track-changes.docx').

Reviewer #2: *Throughout the manuscript (including figures), the authors should correct the word font of mycoplasma taxonomic names. For example, use "Candidatus [italic] Mycoplasma salmoninae [regular] mykiss [regular]" and Mycoplasma iowae [italic] 695 [regular]. Strain names should never be italic.*

=> Thanks to the reviewer for pointing out this. This has been corrected throughout.

Reviewer #2:

Some minor comments:

Line 60. Phylogenetic placement is very clear at the time even without genomes.

=> Thanks to the reviewer for pointing out this. This has been reworded (see lines 58-59 in 'COMMSBIO-20-3080B_MS w track-changes.docx').

Line 89. Here must be specified to pathogenic or symbiotic mycoplasmas as most of environmental mycoplasma genomes were between 1-2 Mbp.

=> This has now been specified as relating to host-associated (both pathogenic and symbiotic) *Mycoplasma* species (see line 93-94 in 'COMMSBIO-20-3080B_MS w track-changes.docx').

Suppl Figs 1 and 2. I have to complain the tiny words on axes.

=> Thanks to the reviewer for pointing out this. This has been improved.

Lines 139-141. Please move to discussion.

=> Thanks to the reviewer for pointing out this. This has been corrected (see lines 364-368 in 'COMMSBIO-20-3080B_MS w track-changes.docx').

Lines 174-175. First I suggest to move the sentence to discussion. All symbionts come from environments. There are about 0.1% of Tenericutes in marine waters. This is enough for acquisition from environment unless the authors show evidence of vertical transmission. I think Fig S5a can be a tree for 16S genes that include the closest relatives from fish or environment if the authors want to show the source of the symbiotic mycoplasma.

=> Thanks to the reviewer for pointing out this. We do agree with reviewer that this would be a great way to scratch the surface of the origin of the salmonid associated *Mycoplasma*. Unfortunately, only one of the MAGs contain a successful assembled ribosomal 16S gene, which is an often limitation of current MAG technology, which would such tree highly difficult to make.

<https://www.nature.com/articles/s41564-017-0012-7>

<https://academic.oup.com/bioinformatics/article/31/12/i35/215357?login=true>

Furthermore, a recent comprehensive study of uncultured Tenericutes (Wang et al. 2020, BMC Genomics), as the reviewer previously has referred to, contains a great overview of phylogeny of environmental Tenericutes and known *Mycoplasma*, which indicates that *Mycoplasma* harboured in the *Suis* clade (including Clade VI with salmonid related *Mycoplasma*) do not originate from environmental Tenericutes (e.g. sediments, waste water, or sewage water), though closest relative to the *Suis* clade is one uncultured Tenericutes originate from marine animals.

<https://bmcgenomics.biomedcentral.com/articles/10.1186/s12864-020-06807-4#Sec15>

<https://bmcgenomics.biomedcentral.com/articles/10.1186/s12864-020-06807-4/figures/1>

Lines 191-203. There is an existing nomenclature for mycoplasma groups. Please refer to Wang et al BMC genomics and Oshima J Mol Evol 2007 65: 714. All the genomes in the Fig 3 have been included in different known groups.

=> Thanks to the reviewer for pointing out this. Citations have been included (see lines **218-219** in 'COMMSBIO-20-3080B_MS w track-changes.docx').

Line 263. Do they lack amino acid synthesis genes? This is a remarkable signal of host dependence.

=> Thanks to the reviewer for pointing out this.

Lines 126, 454, 458, 460, 523,529. Were or are should be was or is! Please check carefully all other gramma problems.

=> Thanks to the reviewer for pointing out this. This has been corrected throughout the paper.

Line 465. Gramma error, rephrase please. A genome cannot contain a bacterium.

=> Thanks to the reviewer for pointing out this. This has been corrected (see lines **561-562** in 'COMMSBIO-20-3080B_MS w track-changes.docx').

Line 473. Put a reference for substitution saturation.

=> Thanks to the reviewer for pointing out this. Citation has been added (see lines **575** in 'COMMSBIO-20-3080B_MS w track-changes.docx').

Line 512. minbit heuristic of 0.5 was implemented is better?

=> Thanks to the reviewer for pointing out this. We agree with reviewer and this should be in accordance with newest version manuscript.

Line 528. Modify qvalue to q-value.

=> Thanks to the reviewer for pointing out this. The line refers to R-package "qvalue".

<https://bioconductor.org/packages/release/bioc/html/qvalue.html>

Figure 3 contains too many genomes and results. I am afraid the words are too tiny to be visualized. I suggest the authors to keep only useful information. ANI color bar is missing.

=> Thanks to the reviewer for pointing out this. We have minimised redundant information.